# Construction of a Winter Wheat Comprehensive Growth Monitoring Index Based on a Fuzzy Degree Comprehensive Evaluation Model of Multispectral UAV Data

**DOI:** 10.3390/s23198089

**Published:** 2023-09-26

**Authors:** Jing Yu, Shiwen Zhang, Yanhai Zhang, Ruixin Hu, Abubakar Sadiq Lawi

**Affiliations:** 1School of Spatial Informatics and Geomatics Engineering, Anhui University of Science and Technology, Huainan 232001, China; jingyaust@163.com; 2School of Earth and Environment, Anhui University of Science and Technology, Huainan 232001, China; hrx19960429@163.com (R.H.); abubakarsadiq830@yahoo.com (A.S.L.); 3Huaibei Mining (Group) Co., Ltd., Huaibei 235001, China; zyh_1234560815@163.com

**Keywords:** comprehensive growth, UAV, fuzzy degree comprehensive evaluation, machine learning

## Abstract

Realizing real-time and rapid monitoring of crop growth is crucial for providing an objective basis for agricultural production. To enhance the accuracy and comprehensiveness of monitoring winter wheat growth, comprehensive growth indicators are constructed using measurements of above-ground biomass, leaf chlorophyll content and water content of winter wheat taken on the ground. This construction is achieved through the utilization of the entropy weight method (EWM) and fuzzy comprehensive evaluation (FCE) model. Additionally, a correlation analysis is performed with the selected vegetation indexes (VIs). Then, using unmanned aerial vehicle (UAV) multispectral orthophotos to construct VIs and extract texture features (TFs), the aim is to explore the potential of combining the two as input variables to improve the accuracy of estimating the comprehensive growth indicators of winter wheat. Finally, we develop comprehensive growth indicator inversion models based on four machine learning algorithms: random forest (RF); partial least squares (PLS); extreme learning machine (ELM); and particle swarm optimization extreme learning machine (PSO-ELM), and the optimal model is selected by comparing the accuracy evaluation indexes of the model. The results show that: (1) The correlation among the comprehensive growth indicators (CGIs) constructed by EWM (CGI_ewm_) and FCE (CGI_fce_) and VIs are all improved to different degrees compared with the single indicators, among which the correlation between CGI_fce_ and most of the VIs is larger. (2) The inclusion of TFs has a positive impact on the performance of the comprehensive growth indicator inversion model. Specifically, the inversion model based on ELM exhibits the most significant improvement in accuracy. The coefficient of determination (*R*^2^) values of ELM-CGI_ewm_ and ELM- CGI_fce_ increased by 20.83% and 20.37%, respectively. (3) The CGI_fce_ inversion model constructed by VIs and TFs as input variables and based on the ELM algorithm is the best inversion model (ELM-CGI_fce_), with *R*^2^ reaching 0.65. Particle swarm optimization (PSO) is used to optimize the ELM-CGI_fce_ (PSO-ELM-CGI_fce_), and the precision is significantly improved compared with that before optimization, with *R*^2^ reaching 0.84. The results of the study can provide a favorable reference for regional winter wheat growth monitoring.

## 1. Introduction

Rapid, accurate and real-time monitoring of crop growth is of great significance in guiding field crop management and crop quality assurance. Traditional methods of monitoring crop growth, including manual sampling and qualitative identification by human eyes, are not only time-consuming and labor-intensive but can also cause damage to the original crop growth [1]. At present, remote sensing technology has been widely used in crop growth monitoring by virtue of its advantages of fast, non-destructive and large monitoring area [2].

Remote sensing technology allows for the acquisition of a wide range of remote sensing data through ground-based, airborne and satellite platforms with different sensors. The ground sensor used for crop monitoring is mainly a canopy optical instrument, which is simple to operate and can realize rapid access. However, the monitoring range of canopy optical instruments is small, requires more labor, and makes it more difficult to obtain high-throughput crop phenotypic information [3]. A satellite remote sensing platform is used for acquiring large-scale remote sensing data. In addition, it has limitations in achieving real-time monitoring of crops with multiple temporal phases for precise operational management due to revisit cycles, weather conditions and resolution. In contrast, unmanned aerial vehicle (UAV) remote sensing monitoring technology has the advantages of high stability, flexible take-off and landing and low cost. It can be equipped with multi-spectral, hyperspectral and other sensors to accurately obtain high-resolution multi-temporal remote sensing image data of crops in real-time, which can account for the deficiency of satellite remote sensing monitoring technology. UAV remote sensing technology has found extensive applications in fields such as agriculture and environmental monitoring [4]. Hasan et al. conducted a study demonstrating the significant potential of UAVs equipped with RGB imagery for estimating the leaf area index of winter wheat during the nodulation stage in mono-crop fields [5]. Wang et al. showed that UAV multispectral imagery could accurately estimate high-throughput phenotypes of other crops in the field, such as the nitrogen content of rice leaves [6]. Zhu et al. reached the conclusion that images captured by a hyperspectral camera mounted on the UAV can effectively monitor the leaf chlorophyll content of maize and wheat with a high level of accuracy [7].

Previous studies on crop monitoring using UAV remote sensing imagery have shown that there is a good correlation among the spectral bands of the imagery, as well as the vegetation indexes (VIs) derived from linear or non-linear combinations of these bands and crop growth indicators. However, it has also been observed that the correlation decreases when the crops are either sparse or overly dense [8]. To address this issue, Sapkota et al. extracted plant height information from the digital surface model obtained from drones and utilized vegetation indices to estimate maize biomass [9]. Dilmurat et al. collected both LIDAR and hyperspectral data from drones and used the LIDAR point cloud to extract canopy structure information, thereby improving the accuracy of maize yield prediction [10]. However, these methods increase the cost and complexity of image acquisition and processing. Texture information extraction is a rapid image processing technique. Texture information can describe the spatial variations in grayscale within an image [11]. Crops with different growth form canopy and horizontal structure differences leading to changes in image texture information [12]. Texture information has been used in fields such as vegetation classification and biomass estimation of forests. There have also been studies that utilize texture features (TFs) for monitoring the single growth indicator of crops. Yue et al. indicated that combining the vegetation index with texture characteristics could improve the prediction accuracy of the above-ground biomass of winter wheat with high canopy coverage [13]. Biomass, chlorophyll content and leaf water content are crucial indicators for monitoring crop growth.

In recent years, there have been studies using UAV remote sensing technology to monitor these growth indicators, and significant progress has been made. Yue et al. used the UAV digital camera and an imaging spectrometer to obtain the remote sensing spectral index of winter wheat, and then the images were subjected to wavelet decomposition and continuous wavelet variations, and finally, the combination of the acquired variables improved the biomass estimation accuracy of winter wheat [14]. Huang et al. utilized a UAV to capture RGB images of rapeseed and combined the R, G and B bands to extract different feature parameters for constructing various inversion models. This study provided a viable reference for estimating chlorophyll content in rapeseed [15]. Ndlovu et al. showed that near-infrared and red-edge-derived spectral variables extracted from UAV multispectral imagery are crucial for characterizing maize moisture indicators [16]. Most studies evaluate crop growth using a single growth indicator, which overlooks the influence of other factors and leads to a one-sided characterization of crop growth. Currently, many studies construct comprehensive growth indicators by assigning equal weights to different indicators. However, this approach overlooks the contribution rates of each indicator and subjective biases in the weighting process. Considering the contribution rate and combining various indicators to construct a comprehensive growth indicator to monitor crop population information more comprehensively needs to be studied in depth [17]. Fuzzy comprehensive evaluation (FCE) is a non-linear multi-objective evaluation method that employs the theory of fuzzy mathematical membership degree to address the uncertainty and inaccuracy in quantitative problems. FCE has found extensive applications in the fields of system evaluation and environmental assessment. Sun et al. utilized FCE to calculate the comprehensive advantages of multiple agronomic trait evaluations, constructed a comprehensive lodging evaluation index and effectively monitored the severity of corn lodging using UAV multispectral imagery [18]. Wang et al. combined the analytic hierarchy process (AHP) with FCE to optimize the irrigation and fertilization quantities for potatoes [19].

In this study, UAV remote sensing is utilized to obtain multispectral ortho imagery of the study area. The research focuses on the above-ground biomass (AGB), leaf chlorophyll content and leaf water content (LWC) of winter wheat. This study employs two methods, namely the entropy weight method (EWM) and the FCE model, to construct comprehensive growth indicators for winter wheat. The input variables include VIs, TFs and a combination of VIs and TFs. Four inversion models are developed, specifically random forest (RF); partial least squares (PLS); extreme learning machine (ELM); and particle swarm optimization extreme learning machine (PSO-ELM), to estimate the comprehensive growth indicators of winter wheat. Finally, the best inversion model is selected based on accuracy comparison in order to investigate the effectiveness of UAV multispectral imagery in monitoring the comprehensive growth indicator of winter wheat. This study provides valuable insights into the application of UAV remote sensing technology in monitoring and managing the growth of winter wheat.

## 2. Materials and Methods

### 2.1. Overview of the Study Area

The study area is located in Suixi County, Huaibei City, Anhui Province, China, with geographic coordinates ranging from 116°24′ to 117°03′ E, 33°16′ to 34°10′ N, and an area of about 0.24 km^2^. The study area has a monsoon warm-temperate semi-humid climate, with mild rain in spring and fall, hot and rainy summer and cold and windy winter. The average annual temperature is 17 °C, the average wind speed is 2.2 m/s, the average precipitation is 737 mm, rainfall is mostly concentrated in June–August and the annual frost-free period is about 218 days.

Crops in the study area are planted in the rotation of winter wheat and summer corn, and the irrigation method is rain-fed. The location of the study area and the distribution of sampling points are shown in Figure 1. After field research, the study area has some land collapse and land destruction due to long-term underground coal mining. Therefore, its elevation shows a gradual decline towards the center collapse area. In order to repair the ecological safety problems caused by coal mining, a series of land reclamation measures have been carried out in the collapsed area.

### 2.2. Data Acquisition

#### 2.2.1. UAV Multispectral Imagery

In this study, the DJ-Innovations (DJI) Phantom 4 multispectral drone real-time kinematic (RTK) was used with an integrated visible light sensor and 5 multi-spectral sensors (blue, green, red, red-edge and near-infrared). The drone also had a built-in global positioning system (GPS) and inertial measurement unit (IMU) system. The parameters of the five bands of the sensor are shown in Table 1.

To avoid bad weather such as rain, snow and high winds, we chose to conduct aerial photography operations when the weather was clear and cloudless and the wind was light. We conducted a survey of the geographical conditions of the study area and the surrounding structures to ensure avoidance of interference from objects such as high-voltage power lines and tall trees during the drone flight. The collection of multispectral imagery by the drone took place on 10 January 2021, with a flight time between 11:00 a.m. and 2:00 p.m. Before takeoff, the drone was manually controlled to fly approximately 2.5 m directly above the calibration whiteboard. The camera was set to single-shot mode to capture images of the standard whiteboard. The whiteboard model used in this study was an SRT-99-100 with a diffuse reflectance of 99% and a size of 25.4 cm × 25.4 cm. The drone’s flight mode followed a pre-planned S-shaped flight path. The heading overlap was set to 75% and the side lap was set to 60%. The drone flew at a height of 100 m with the sensor lens facing vertically downward. The camera was set to capture images at equal time intervals, with a time interval of 3 s. Photographs of the operational site for UAV field data collection are shown in Figure 2.

After acquiring the image, Pix4D-mapper 4.5.6 software was used to assemble the image to obtain the orthographic image of the study area. Using ArcMap 10.6 software and referencing the orthoimage as the base image, 30 reference points were uniformly selected for geometric correction on each of the five single-band images. The geometric correction error was within 0.5 pixels. Afterward, the digital number (DN) values of the whiteboard were obtained for each band, and the average DN values of the five bands were calculated. The multispectral images were then radiometrically corrected using the Band Math functionality in ENVI 5.3 software. The pre-processed UAV multispectral images were acquired after the above processing with a resolution of 0.05697 m. Finally, the pre-processed multispectral images, as well as the GPS point locations of the sampling points, were imported into the ENVI 5.3 software. An 18 × 18 (pixel) image centered on the sampling point, corresponding to the sample quadrat size and image resolution, was cropped as the region of interest. The average reflectance of each band within each sample quadrat was extracted.

#### 2.2.2. Ground Data Acquisition

Sample point layout was conducted before the UAV aerial survey. After the image collection was completed, the data collection for winter wheat growth indicators was immediately carried out. The sample point layout in the study area adopted the method of uniform distribution, and the positions of the sample quadrats were recorded using the GPS. Fifty-four sample quadrats were laid out in the area, each with a size of 1 m × 1 m. The sampling and testing methods for ground-based measurements were as follows.

(1)Above-ground biomass (AGB)

The five-point sampling method was applied to the 54 sample quadrats laid out in this study, and the locations of the 54 sample quadrats in the study area are shown in Figure 1. A representative number of 20 winter wheat plants were selected from each sample quadrat. The plants were removed flush with the ground and the leaves and stems were separated and stored in sealed bags. The temperature of the laboratory oven was set at 105 °C for 30 min of greening treatment, and then the plants were placed in an oven at 80 °C for 24–48 h until a constant dry biomass was obtained [20,21].

(2)Leaf chlorophyll content (SPAD)

The SPAD-502 chlorophyll instrument determines plant leaf SPAD values. Its measured leaf SPAD values are highly significantly correlated with leaf chlorophyll content [7]. Therefore, this instrument was used in this study to determine the chlorophyll content of winter wheat leaves. Five winter wheat plants were selected in each sample quadrat using the five-point sampling method. Then, the SPAD values of the leaf tip, leaf center and leaf base of each winter wheat plant were measured, and the average of the SPAD values of these three parts was calculated as the SPAD value of the winter wheat plant. Finally, the average of the SPAD values of the five winter wheat plants was taken as the chlorophyll content of the winter wheat in the sample quadrat [22].

(3)Leaf water content (LWC)

After leaf stem separation, the fresh leaf weight of the winter wheat plant sample was weighed. Dry leaf dry weight was obtained after drying. The leaf water content was calculated according to Equation (1) [23]:(1)$$LWC=\frac{FW-DW}{FW}$$
where *FW* is the fresh weight of the leaf; *DW* is the dry weight of the leaf.

### 2.3. Methods

The research framework is shown in Figure 3. The following work is illustrated in the figure:(1)Acquisition and preprocessing of UAV multispectral data in the study area, which is in Section 2.2.1 of the article. Based on the preprocessed multispectral images, vegetation indices (VIs) and texture features (TFs) are extracted as input feature variables for the inverse model of comprehensive growth indicators;(2)Construction of two kinds of winter wheat comprehensive growth indicators. The above-ground biomass (AGB), leaf chlorophyll content (SPAD) and leaf water content (LWC) of winter wheat are measured using field samples, and CGI_ewm_ and CGI_fce_ are constructed based on the entropy weight method (EWM) and fuzzy degree comprehensive evaluation model (FCE), respectively;(3)Construction and validation of the inversion model of comprehensive growth indexes. The features were categorized into three feature groups: vegetation index; texture features; and the combination of the two, which are used as input variables to construct the inversion models of CGI_ewm_ and CGI_fce_, with ①, ② and ③ indicating the variable groupings and the order of model construction, respectively. The four regression algorithms selected in this study are partial least squares (PLS), random forest (RF), extreme learning machine (ELM) and particle swarm optimization extreme learning machine (PSO-ELM) for model construction. The accuracy of the inverse model is verified by *R*^2^ and nRMSE.

#### 2.3.1. Vegetation Indexes (VIs) Construction

Different vegetation species and different growth conditions cause different absorption and scattering effects of incident light in different wavelength bands to enable the characterization of the spectral response between vegetation and incident light [24]. Vegetation indexes (VIs) are calculated by combining the reflectance of different wavelength bands in a linear or nonlinear manner. Therefore, compared with a single spectral reflectance, the VI has a more sensitive response to vegetation growth and can weaken the influence of environmental background on the spectral reflectance of the canopy [25,26]. Based on the results of previous studies, 12 VIs were selected for vegetation growth monitoring (Table 2).

#### 2.3.2. Texture Features (TFs) Extraction

Texture refers to the repeated sequence patterns and regular arrangement and distribution in an image, which can reflect the spatial distribution information of the image grayscale [38]. In this study, one of the most widely used texture extraction techniques, the gray-level co-occurrence matrix method, was used to obtain the texture parameters of multispectral images. This method is based on the comprehensive acquisition of local features and arrangement rules of image pixels [39]. In this study, eight texture features in R-band and G-band were extracted using ENVI 5.3 software. The bit-depth of both the R-band and G-band was 8-bits. The formula for each texture feature is shown in Table 3.

#### 2.3.3. Entropy Weight Method and Fuzzy Comprehensive Evaluation Model

(1)Entropy weight method (EWM)

Entropy can be used in information theory to measure the amount of useful information provided by various indicators. When using entropy to calculate the weights of indicators, the magnitude of entropy weight corresponds to the amount of information that the selected indicators provide for evaluating the growth status of winter wheat. The larger the entropy weight, the greater the amount of information [40,41]. Therefore, the EWM was used to assign weights to each indicator in this study. The steps for calculating the weights are as follows:

(i) Normalize the original data to obtain a standard matrix:(2)$$X^{\prime }=\left\{x_{ij}^{\prime }\right\}$$
where $x_{ij}^{\prime }$ represents the normalized value of the *j*th sample quadrat for the *i*th index;

(ii) Calculate the entropy of the indicator *e_i_*. Calculate the information entropy of the three longevity indicators: AGB; SPAD; and LWC. *e_i_* denotes the information entropy of each longevity indicator of *i*. The smaller the value of information entropy and the larger the value of entropy weight, the more informative the indicator is for the constructed comprehensive growth indicator:(3)$$e_{i}=-k\sum_{j=1}^{m}x_{ij}^{\prime }\ln x_{ij}^{\prime }\;\;\;\;j=1,2,\cdots ,m$$
(4)k=1/lnn
where *k* is the standardized coefficient, *m* is the number of sample quadrats and *n* is the number of indicators;

(iii) The weight of each indicator is: $w_{i}=\left(1-e_{i}\right)/\left(n-\sum_{i=1}^{n}e_{i}\right)$.

The weight vector *W* = [*w*_1_, *w*_2_, *w*_3_] is obtained, and *w*_1_, *w*_2_ and *w*_3_ are the weights of each indicator, respectively.

The comprehensive index constructed by the entropy weight method (CGI_ewm_) is represented as Equation (5):(5)$${\mathrm{CGI}}_{\mathrm{ewm}}=w_{1}x_{1}^{\prime }+w_{2}x_{2}^{\prime }+w_{3}x_{3}^{\prime }$$
where, $x_{1}^{\prime }$, $x_{2}^{\prime }$ and $x_{3}^{\prime }$ are the respective normalized values of the indicators.

(2)Fuzzy comprehension evaluation method (FCE)

The fuzzy comprehension evaluation (FCE) method is based on fuzzy mathematics. It utilizes the theory of membership degrees in fuzzy mathematics to transform qualitative evaluations into quantitative ones, enabling a comprehensive evaluation of objects that are influenced by multiple factors [42,43]. The above-ground biomass, leaf chlorophyll content and leaf water content of winter wheat were measured in this study. The weights of the EWM combined with the fuzzy degree comprehensive evaluation model were used to construct a comprehensive indicator to quantify the winter wheat growth information. The steps of constructing the comprehensive fuzzy evaluation model are as follows:

(i) Determine the set of factors and weight vector. The set of factors composed of the evaluation object’s elements is referred to as the factor set. If there are *n* evaluation factors, the factor set can be represented as follows: *U* = {*u*_1_, *u*_2_, …, *u_n_*}. As this study involves three influencing factors, the factor set can be defined as follows: *U* = {*u*_1_, *u*_2_, *u*_3_}. *u*_1_, *u*_2_ and *u*_3_ are winter wheat AGB, SPAD and LWC, respectively.
(6)$$CGI_{\mathrm{fce}}=f\left(AGB,SPAD,LWC\right)$$
where CGI_fce_ is a comprehensive growth indicator constructed by FCE;

(ii) Determine the evaluation level. The collection of evaluation results is called the evaluation set. Based on the actual growth of winter wheat in the study area. The growth indicators are categorized into five evaluation levels: *V* = {*v*_1_, *v*_2_, *v*_3_, *v*_4_, *v*_5_}. The *v*_1_, *v*_2_, *v*_3_, *v*_4_ and *v*_5_ represent poor growth, relatively poor growth, medium growth, relatively good growth and good growth, respectively;

(iii) Determine the fuzziness matrix and membership functions. The fuzzy matrix is denoted as *R_nm_* = {*r_nm_*}, where *r_nm_* represents the evaluation result in the *m*th evaluation set for the *n*th evaluation factor. Based on the defined set of factors and evaluation sets, the fuzzy matrix for this study can be obtained as shown in Equation (7):(7)$$R=\left[\begin{array}{c} r_{11}\\ r_{21}\\ r_{31} \end{array}\begin{array}{c} r_{12}\\ r_{22}\\ r_{32} \end{array}\begin{array}{c} r_{13}\\ r_{23}\\ r_{33} \end{array}\begin{array}{c} r_{14}\\ r_{24}\\ r_{34} \end{array}\begin{array}{c} r_{15}\\ r_{25}\\ r_{35} \end{array}\right]$$

The membership function is utilized in fuzzy set theory to accurately represent and address real-world evaluation problems through the ranking and partitioning of fuzzy concepts. Considering the ambiguity in determining membership based on interval boundaries, this study utilized trapezoidal membership functions to construct the fuzzy degree matrix. The [0,1] was divided equally into 5 intervals corresponding to 5 rating levels. The fuzzy boundary gap between the head and tail was 0.15, and the center point gap was 0.1. The fuzzy boundary centers for adjacent rating levels were 0.2, 0.4, 0.6 and 0.8, respectively, as shown in Figure 4.

(4) Construct a comprehensive evaluation model of fuzziness and assessment values. Multiply the weight vector obtained by the entropy weighting method with the fuzziness matrix to obtain the fuzzy vector *B* of the evaluation set. Quantify the score values to a percentage using the score vector as *F* = [100 75 50 25 0]. *S* represents the evaluation value, which is the final numerical value of various comprehensive growth indicators for winter wheat sample quadrats, as shown in Equations (8) and (9):(8)B=W•R
(9)$$S=B•F^{T}$$
where *W* is the weight vector of the evaluation factors obtained using the EWM.

#### 2.3.4. Machine Learning Algorithms

Using regression analysis, the inversion models of winter wheat comprehensive growth indexes constructed based on EWM and FCE were established, respectively. Four algorithms were selected to construct the inversion models, namely partial least squares (PLS), random forest (RF), extreme learning machine (ELM) and particle swarm optimization extreme learning machine (PSO-ELM) regression algorithms, respectively. In this study, the above algorithms were completed in MATLAB 2022a software for the inversion model of comprehensive growth indicators of winter wheat.

PLS is a multivariate linear regression analysis method that can simplify data structure and analyze the correlation among variables. When building regression models, it can achieve data dimensionality reduction, information synthesis and screening techniques to extract new composite components that best explain the system [44].

RF is an ensemble learning method that combines multiple decision tree algorithms to generate repeated estimations, which solves the problem of multicollinearity among variables. By calculating the importance of variables, RF can be used to select variables and perform complex nonlinear regression [45]. In this study, the RF model was configured with 100 decision trees and a minimum leaf size of 5.

ELM is a feedforward neural network algorithm with a single hidden layer. The model structure consists of an input layer, a hidden layer and an output layer. The input weights and hidden layer have random values and do not require adjustment. Each layer is connected through a feature mapping function, which offers the advantages of efficient and fast learning [46]. Previous research has indicated that ELM demonstrates better predictive accuracy for monitoring vegetation growth phenotypic traits compared to several typical empirical models. After multiple runs, the optimal number of neurons in the hidden layer was determined to be 5, and the activation function was set as the “sigmoid” function.

PSO-ELM regression algorithm combines the PSO algorithm with ELM to address the issue of randomness in the input weights and hidden layer [47]. The PSO optimization algorithm mimics the behavior of a flock of birds searching for food using a stochastic optimization technique. Each bird in the flock is represented as a particle. In the optimization process, each particle searches for the global best position with a certain flying speed. They explore the solution space based on the fitness function and iteratively update their velocity and position by tracking individual and global extreme values. The principle is as follows: In a D-dimensional search space, a population consisting of n particles, denoted as *X* = (*x*_1_, *x*_2_, …, *x_D_*), is present. The position of the *i*th particle is represented by the vector *X_i_* = (*x_i_*_1_, *x_i_*_2_, …, *x_iD_*)*^T^*, *i* = 1, 2, …*n*, while its velocity is represented *V_i_* = (*v_i_*_1_, *v_i_*_2_, …, *v_iD_*)*^T^*, *i* = 1, 2, …*n*. The fitness function *f* (*X_i_*) calculates the fitness value of the particle’s position. Each particle has an individual best value *p_besti_* = (*p_i_*_1_, *p_i_*_2_,…, *p_iD_*), and the entire population has a global best value *g_besti_* = (*g_i_*_1_, *g_i_*_2_,…, *g_iD_*). The particles update their velocity and position iteratively based on Equations (10) and (11).
(10)$$V_{id}^{k+1}=wV_{id}^{k}+c_{1}r_{1}\left(p_{id}^{k}-X_{id}^{k}\right)+c_{2}r_{2}\left(g_{id}^{k}-X_{id}^{k}\right)$$
(11)$$X_{id}^{k+1}=X_{id}^{k}+V_{id}^{k+1}$$
where *k* represents the current iteration count, c1 and *c*_2_ are non-negative constant acceleration factors, *w* is the inertia weight factor, *r*_1_ and *r*_2_ are random uniform numbers between 0 and 1 and $p_{id}^{k}$ and $g_{id}^{k}$ represent the individual best position and global best position of the *i*th particle in the *d*th dimension during the *k*th iteration. The optimization steps are as follows:(1)Initialize the particle swarm. PSO parameters include acceleration constants, inertia weight, particle dimensions, maximum iteration count and population size;(2)Train the ELM algorithm with random input weights and thresholds for each particle to obtain the output weight prediction. The root mean square error calculated from the training samples is used as the particle fitness. Update the individual and global best values based on the higher fitness value. During the iteration process, update the particle’s velocity and position using Equations (10) and (11). Stop the iteration when reaching the maximum iteration count or the best fitness;(3)Obtain the optimal fitness and hidden layer thresholds and input them into the ELM structure to calculate the weight matrix and obtain the prediction results.

After multiple training sessions, the parameters of the particle swarm were determined as follows: the maximum iteration count was set to *k* = 100; the number of particles in the population was set to *n* = 40; the acceleration constant was set to *c*_1_ = 2.8; the inertia weight was set to *c*_2_ = 1.3; and the velocity range was [−1,1].

#### 2.3.5. Evaluation of Model Accuracy

In this study, a random sample of 40 out of 54 data points was selected for model training, while the remaining 14 were used for model validation. The model performance was quantitatively evaluated using the coefficient of determination (*R*^2^) and the normalized root mean square error (nRMSE), as shown in Equations (12) and (13). *R*^2^ represents the degree of fit of the model, while nRMSE indicates the magnitude of the error between predicted and actual values. A higher *R*^2^ and lower nRMSE indicate higher accuracy of the model [48].
(12)$$R^{2}=1-\frac{\sum_{i=1}^{n}{\left(y_{i}-{\hat{y}}_{i}\right)}^{2}}{\sum_{i=1}^{n}{\left(y_{i}-\bar{y}\right)}^{2}}$$
(13)$$\mathrm{nRMSE}=\frac{\sqrt{\frac{1}{n}\sum_{i=1}^{n}{\left(y_{i}-{\hat{y}}_{i}\right)}^{2}}}{\bar{y}}\times 100\%$$
where *y_i_* is the measured value, ${\hat{y}}_{i}$ is the predicted value, $\bar{y}$ is the mean of the measured value and *n* is the number of samples.

## 3. Results and Analysis

### 3.1. Comprehensive Growth Indicator Construction

In this study, the entropy weight method and fuzzy comprehensive evaluation model were used to construct winter wheat comprehensive growth indicators, which are represented by CGI_ewm_ and CGI_fce_, respectively. The result of comprehensive growth index constructed by the entropy weight method is shown in Equation (14). The comprehensive growth indicator values of various methods constructed by the two methods are shown in Figure 5.
(14)$${\mathrm{CGI}}_{\mathrm{ewm}}=0.624u_{1}^{\prime }+0.093u_{2}^{\prime }+0.283u_{3}^{\prime }$$
where $u_{1}^{\prime }$ represents the normalized above ground biomass of winter wheat, $u_{2}^{\prime }$ represents the normalized leaf water content and $u_{3}^{\prime }$ represents the normalized chlorophyll content of winter wheat.

According to Equation (14), it can be observed that compared to leaf water content and chlorophyll content, the above-ground biomass had the highest contribution rate to the comprehensive growth indicator, with a weight of 0.624. The chlorophyll content followed next with a weight of 0.283, while leaf water content had the lowest contribution rate with a weight of 0.093. Figure 5 shows that most of the values of CGI_ewm_ were concentrated between 0.4 and 0.8, and most of the values of CGI_fce_ were concentrated between 25 and 65. The figure illustrates that CGI_ewm_ and CGI_fce_ exhibited opposite trends, suggesting that a smaller value of CGI_fce_ indicates better winter wheat growth. The CGI_fce_ values of Sample 3 and Sample 21 in the figure were zero and the values of CGI_ewm_ were high, indicating that the winter wheat in the sample quadrat was growing well.

### 3.2. Correlation Analysis

To analyze the correlation between the composite growth indicators CGI_ewm_ and CGI_fce_ (CGIs) constructed using two different methods and vegetation indexes, we performed a correlation analysis using the Pearson correlation coefficient among AGB, LWC, SPAD, CGI_ewm_, CGI_fce_ and the selected 12 VIs. The results of the analysis are shown in Figure 6.

As can be seen in Figure 6, there were different degrees of correlation between the three single indicators and the VIs. Among them, the correlation between AGB and VIs was generally larger, and the correlation between SPAD and most of the VIs were larger than that of LWC. CGI_ewm_ and CGI_fce_ passed the significance test with the 12 VIs, and the correlation between the two composite indicators and the VIs increased to different degrees compared to the single indicators, and the increase was the most obvious compared to LWC, and all were correlated with the selected VIs up to the 0.01 level of significant correlation. The correlations between the VIs and CGI_fce_ were greater than those of CGI_ewm_, except for RVI, RDVI and EVI2.

### 3.3. Input Variables

To address the issues of limited input features and the problem of spectral feature saturation, this study introduced TFs as additional input variable for the model. The final selection of VIs as input variables for the VI was based on their significant correlation at the 0.01 level with both CGI_ewm_ and CGI_fce_ (CGIs). Performing correlation analysis between the composite indicators and individual bands (Figure 7a) revealed that both the G and R bands showed significant correlation at the 0.01 level with the two constructed composite indicators. Therefore, the texture information of the G and R bands was further analyzed for correlation with the composite indicators (Figure 7b). The TFs that exhibited significant correlation at the 0.01 level with both composite indicators were selected as the final texture feature selection results.

### 3.4. Model Construction

#### 3.4.1. Inversion Model Construction of Comprehensive Indicators Based on the EWM

According to the above method, the final input variables of the vegetation index and texture feature were determined. A CGI_ewm_ inversion model was constructed based on vegetation index, texture feature and the combination of vegetation index and texture feature. The accuracy of the constructed model is shown in Figure 8.

As can be seen from the inversion validation results displayed in Figure 8a,b, when the CGI_ewm_ inversion model was constructed using Vis and TFs as model input variables, respectively, the accuracy of the models constructed based on the ELM algorithm were both the highest. This was followed by the RF algorithm. PLS constructed the model with the lowest accuracy.

Figure 8c presents the results of constructing the CGI_ewm_ inversion model by combining VIs and TFs as input variables using three different machine learning algorithms. From the figure, it can be observed that compared to using VIs feature as input variable, all models showed varying degrees of improvement in accuracy. Among them, the ELM-CGI_ewm_ model exhibited a significant improvement in accuracy.

The inclusion of the texture feature variable improved the *R*^2^ of the ELM-CGI_ewm_ model by 20.83% and reduced the nRMSE by 9.83% compared to using VIs as the input variable.

Comparing the results of constructing the inversion model using three different algorithms, the ELM-CGI_ewm_ model, which combined VIs and TFs, achieved the highest accuracy in predicting the comprehensive growth indicator.

#### 3.4.2. Inversion Model Construction of Comprehensive Indicators Based on the FCE

The FCE was used to construct the comprehensive growth indicator of winter wheat, also with three variable inputs based on PLS, RF and ELM machine learning algorithms to construct the inverse model of the growth indicator. The results of the construction of each model are shown in Figure 9.

Figure 9a,b represents the model validation results of constructing CGI_fce_ inversions with VIs and TFs input variables, respectively. As can be seen from the figures, ELM-CGI_fce_ exhibited optimal model performance compared to both the PLS-CGI_fce_ and RF-CGI_fce_ models.

Figure 9c shows the results of the comprehensive indicator inversion model construction when the combination of VIs and TFs were used as input variables. Except for the PLS-CGI_fce_ model, the accuracy of all inversion models improved compared with only using a single feature as the input variable. The accuracy of the ELM-CGI_fce_ model was the highest compared with the other eight comprehensive indicator inversion models, with an *R*^2^ of 0.65 and an nRMSE of 16.34%. And the *R*^2^ of the ELM-CGI_fce_ model improved by 20.37%, and the nRMSE reduced by 13.32% compared to when only VIs was used as the input variable.

The comparison of the performance of inversion models for the comprehensive growth indicator of winter wheat constructed in different ways revealed that the CGI_fce_ inversion models, built using the RF and ELM algorithms with various feature input variables, exhibited improved accuracy compared to the CGI_ewm_ inversion model. Thus, it can be concluded that the comprehensive growth indicator constructed using the fuzzy comprehensive evaluation model more effectively reflects the growth information of winter wheat in comparison to the entropy weight method.

#### 3.4.3. Construction of the PSO-ELM-CGI_fce_ Inversion Model

Based on the construction results mentioned above, the ELM-CGI_fce_ inversion model, built using a combination of VIs and TFs as input variables, exhibited good performance. However, due to the random generation of initial weights and thresholds in the ELM model structure, it may suffer from the disadvantage of ineffective hidden nodes and insufficient generalization ability.

To overcome this issue, this study introduced the PSO algorithm to optimize the initial input weights and thresholds of the ELM-CGI_fce_ model, aiming to achieve a higher accuracy inversion model for CGI_fce_. The fitting effect of the PSO-ELM-CGI_fce_ inversion model on the validation samples is shown in Figure 10.

As can be seen from Figure 10a, the CGI_fce_ inversion model *R*^2^ constructed with PSO-ELM increased from 0.65 to 0.84, with an increase of 29.23% and a decrease of 31.82% in nRMSE.

As can be seen from Figure 10b, compared with the scatterplot of the PSO-ELM-CGI_fce_ model before optimization, most points were close to the 1:1 line after optimization of the algorithm, and the fitting line of the prediction point was closer to 1:1 line, indicating that after processing by the optimization algorithm, the difference between the predicted value and the measured value was reduced, and the PSO algorithm improved the prediction accuracy of the ELM-CGI_fce_ model. At the same time, from the scatterplot dispersion degree and fitting line of the measured and predicted values of the training set, it can be seen that the accuracy of the scatterplot was relatively high and similar to that of the test set, indicating that the model has strong generalization ability.

## 4. Discussion

### 4.1. Combination of VIs and TFs as Input Variables

A UAV multispectral remote sensing image interpretation technique can provide an efficient and reliable method for monitoring the growth of winter wheat. Many studies have focused on extracting spectral information to monitor crop growth. However, spectral information tends to saturate when vegetation canopy coverage is high and is easily affected by soil background reflectance in sparse plant areas. Additionally, previous research has shown that the texture information derived from UAV images can accurately predict winter wheat biomass [49,50]. Therefore, in this study, we introduced texture features based on vegetation indexes to construct an inversion model for comprehensive growth indicators of winter wheat that includes biomass. Based on the results of the correlation analysis, the TFs of R and G band were used in this study. Xu et al. found that the TFs of different bands have different effects on rice biomass monitoring, and that the red and green bands have advantages in characterizing texture information [51]. Taking into account that winter wheat is in the winter period and the plants are relatively sparse, the extraction of reflectance information is influenced by the soil background. The green band can reflect the spectral reflectance of background objects, indicating that considering texture features in the red and green bands is more reasonable. When the combination of vegetation index and texture features was used as input variables, the accuracy was significantly improved, except for the PLS-CGI_fce_ model. Among them, the ELM-CGI_fce_ model had a larger accuracy improvement. Compared to using VIs as input variables, *R*^2^ was improved by 20.37%. It indicates that combining texture features with the vegetation index can improve the accuracy of the CGIs prediction. The reason is that image texture can represent the phenotypic information of crops, specifically the differences in field growth details. By incorporating both spectral and texture information, the limitations of relying solely on spectral data and its saturation can be overcome effectively. Additionally, texture information proves to be more stable compared to vegetation indices, even in the presence of noise and the influence of soil background, effectively diminishing the interference caused by weather conditions and soil background [52,53,54].

### 4.2. Comprehensive Growth Indicators Construction

A single growth index can only reflect the growth characteristics of winter wheat in a single aspect, such as morphological structure, physiology and biochemistry, and cannot comprehensively monitor winter wheat growth [55]. Therefore, in this study, EMW and FCE were used to construct CGI_ewm_ and CGI_fce_. Correlation analysis was conducted between two comprehensive indicators and VIs. The results showed that compared with three single indicators, both comprehensive growth indicators had corresponding improvements in correlation. This indicates that CGI_ewm_ and CGI_fce_ contain more information about winter wheat growth, and the selected VIs have a better response relationship with them. Among them, CGI_fce_ has a higher correlation with most VIs, indicating that the comprehensive growth indicator constructed by the fuzzy comprehensive evaluation model can more comprehensively respond to winter wheat growth [56].

The growth of winter wheat is influenced by multiple factors. However, evaluating the growth of winter wheat based on single indicator, such as biomass, plant water content and chlorophyll content, can be subjective and imprecise. To tackle the uncertainties, the FCE can be employed. Additionally, the EWM can objectively determine the weights of different indicators by considering their information entropy. In this study, we developed an FCE that combines the strengths of both approaches, making it more advantageous compared to the EWM.

### 4.3. Inversion Model Construction of Winter Wheat CGIs

The RF, PLS and ELM algorithms were selected to construct regression models, and the results showed that ELM modeling results have the best effect. Maimaitijiang et al. found that the ELM algorithm showed the best performance in estimating crop growth traits [57], which is consistent with the results of this study. The accuracy of the models constructed by the PLS algorithm was lower than the RF and ELM. The reason for this is that there is a nonlinear relationship among Vis, TFs and CGIs. The PLS algorithm used to describe the linear relationship to build the CGI inversion model is deficient [58]. On the other hand, RF and ELM use nonlinear functions to establish the functional relationship between input variables and output variables, which can explain the correlation among Vis, TFs and CGIs more accurately. The RF algorithm is suitable for prediction problems with large volumes of data, and it tends to experience overfitting issues when applied to small data volumes. This leads to poorer results in the construction of its inverse model compared to the ELM algorithm [59,60]. The ELM algorithm has poor generalization ability due to the randomness of input weights and hidden layer thresholds. The PSO algorithm can systematically optimize the input weights and hidden layer thresholds through the stochastic optimization technique of the population [61]. Therefore, from the results of model construction in this study, the accuracy of the PSO-ELM-CGI_fce_ model was significantly improved compared with that before optimization. This study specifically focused on the growth indicators of winter wheat during the overwintering period. However, since winter wheat crops exhibit variations in phenotypic traits throughout different growth stages, future research can delve into and refine monitoring methods for comprehensive growth indicators of winter wheat across multiple growth stages.

## 5. Conclusions

(1)The biomass, leaf chlorophyll content and leaf water content of winter wheat were used to construct the CGIs (CGI_ewm_, CGI_fce_) by EWM and FCE. According to Pearson correlation analysis with VIs, CGI_ewm_ and CGI_fce_ are significantly correlated with each other. The correlation between CGI_fce_ and most VIs is greater than that of CGI_ewm_, and the CGIs of winter wheat constructed by the two methods contain more growth information than the single index. CGI_fce_ has a better response relationship with the selected Vis;(2)When constructing the CGI_ewm_ inversion model, the model accuracy constructed by the RF, PLS and ELM algorithms is improved after introducing TFs as model input variables based on the VIs, and the *R*^2^ is 0.47, 0.51 and 0.58, respectively, of which the ELM is improved the most, with the *R*^2^ improved by 20.83%, and the nRMSE reduced by 9.83%. When constructing the CGI_fce_ inversion model, the accuracy of all algorithms, except PLS, is improved accordingly with the introduction of TFs. The ELM-CGI_fce_ of winter wheat can better reflect the growth of winter wheat in the study area, with *R*^2^ of 0.65 and nRMSE of 16.34%. The combination of VIs and TFs effectively improves the inversion accuracy of the comprehensive growth indicators;(3)After optimizing the ELM-CGI_fce_ model of winter wheat growth by PSO, the prediction accuracy of the model is significantly improved. The *R*^2^ increased from 0.65 to 0.84, which is 29.23% higher, and nRMSE reduced by 31.82%. The PSO algorithm optimizes the parameters of the ELM algorithm, and PSO-ELM-CGI_fce_ more accurately estimates the CGI_fce_ of winter wheat.

In this study, the EWM and FCE were applied to the construction of a comprehensive growth indicator of winter wheat. A variety of algorithms was used to construct the inverse model of the comprehensive growth indicator by combining the vegetation index and texture parameters. The results of this study can provide a valuable reference for monitoring the growth of winter wheat and other field crops.

## Figures and Tables

**Figure 1 sensors-23-08089-f001:**
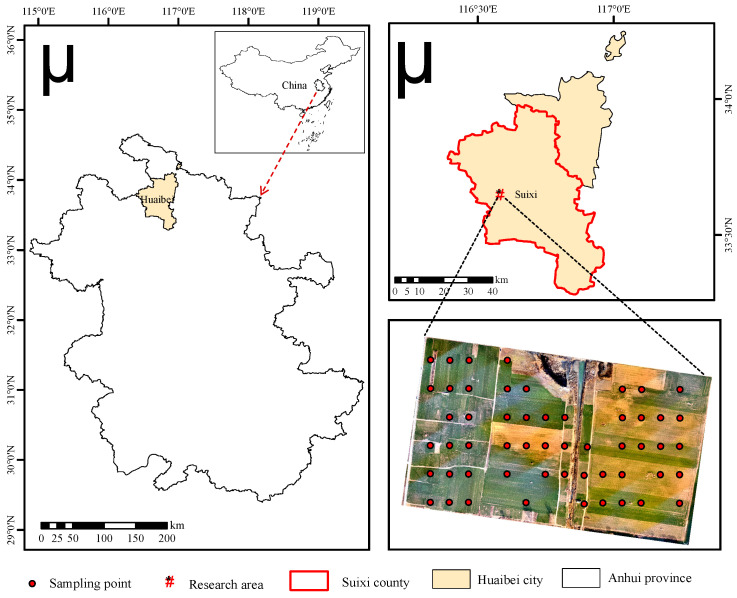
Location of the study area and distribution of sampling points.

**Figure 2 sensors-23-08089-f002:**
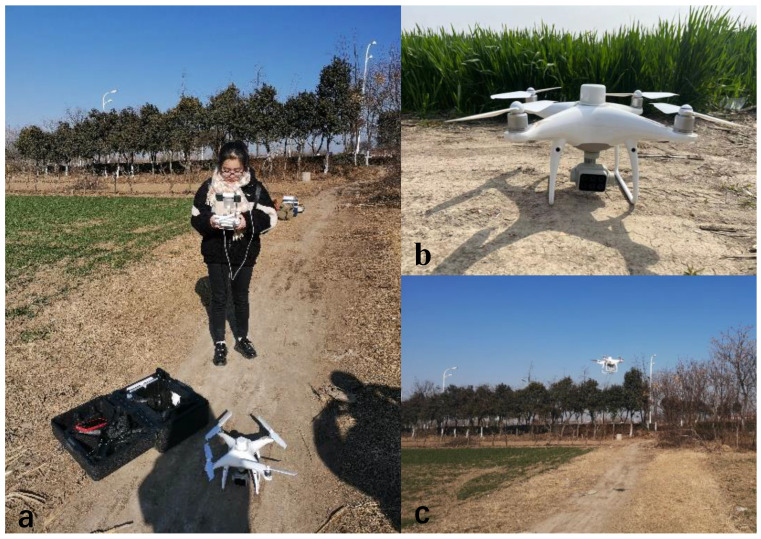
Photographs of drone field sites. UAV operation (**a**), UAV (**b**), UAV flight (**c**).

**Figure 3 sensors-23-08089-f003:**
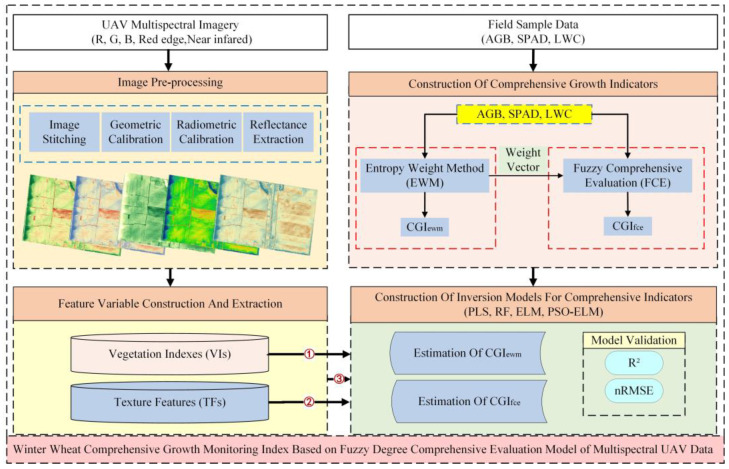
Research framework. Note: AGB stands for above-ground biomass; SPAD indicates leaf chlorophyll content; LWC stands for leaf water content; ①: VIs used as input variables; ②: TFs used as input variables; ③: a combination of VIs and TFs used as an input variable.

**Figure 4 sensors-23-08089-f004:**
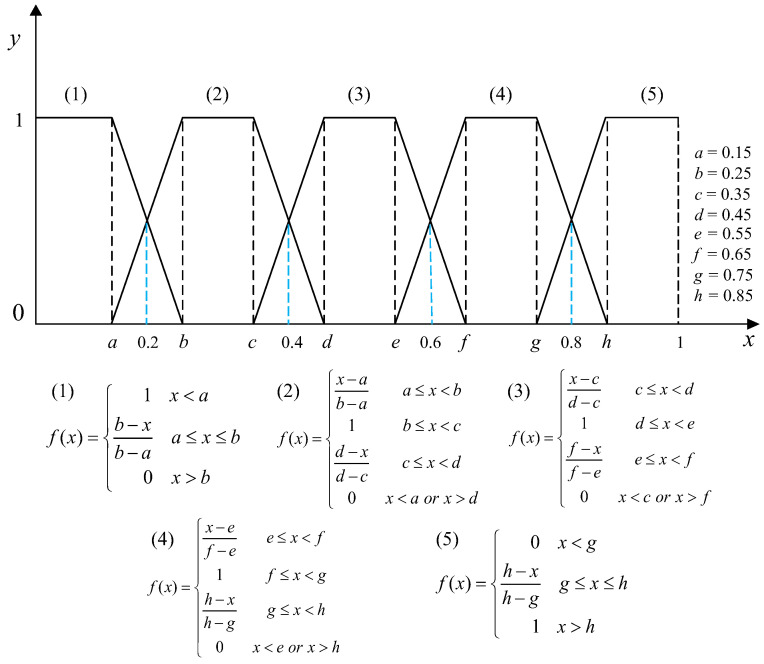
Membership function graphs.

**Figure 5 sensors-23-08089-f005:**
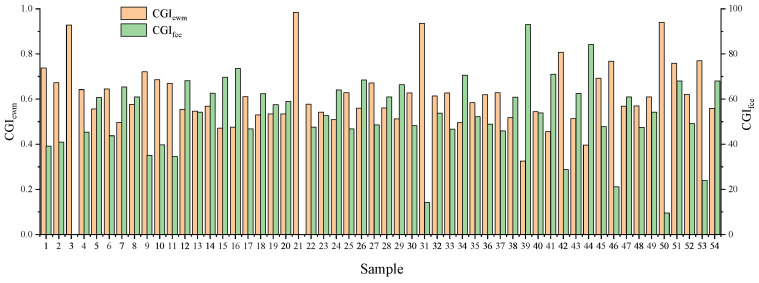
The results of constructing the comprehensive growth index.

**Figure 6 sensors-23-08089-f006:**
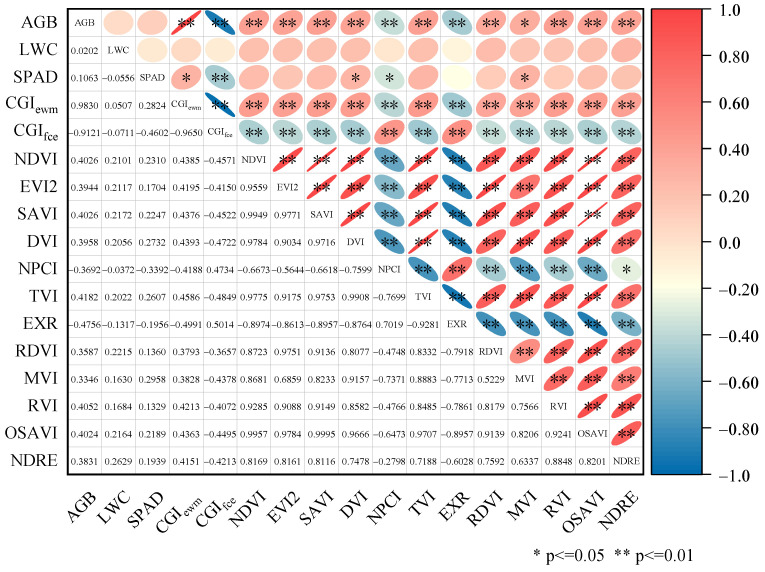
Correlation between growth indicators and vegetation indexes. Note: * indicates significant correlation at the 0.05 level; ** indicates significant correlation at the 0.01 level; red color indicates positive correlation and blue color indicates negative correlation; right tilted ellipse indicates positive correlation and left tilted ellipse indicates negative correlation; flatter ellipse indicates higher correlation and wider ellipse indicates lower correlation.

**Figure 7 sensors-23-08089-f007:**
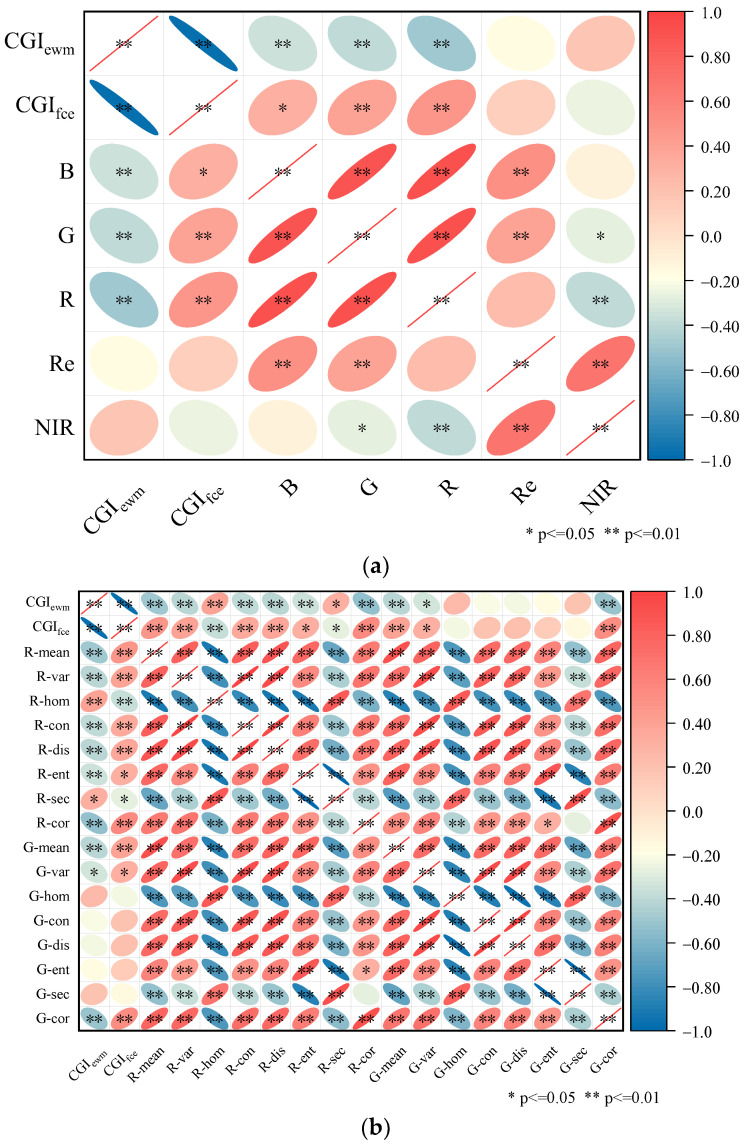
Texture feature correlation analysis. Note: * indicates significant correlation at the 0.05 level; ** indicates significant correlation at the 0.01 level; red color indicates positive correlation and blue color indicates negative correlation; right tilted ellipse indicates positive correlation and left tilted ellipse indicates negative correlation; flatter ellipse indicates higher correlation and wider ellipse indicates lower correlation. (**a**) Correlation between CGIs and single bands. (**b**) Correlation between CGIs and TFs.

**Figure 8 sensors-23-08089-f008:**
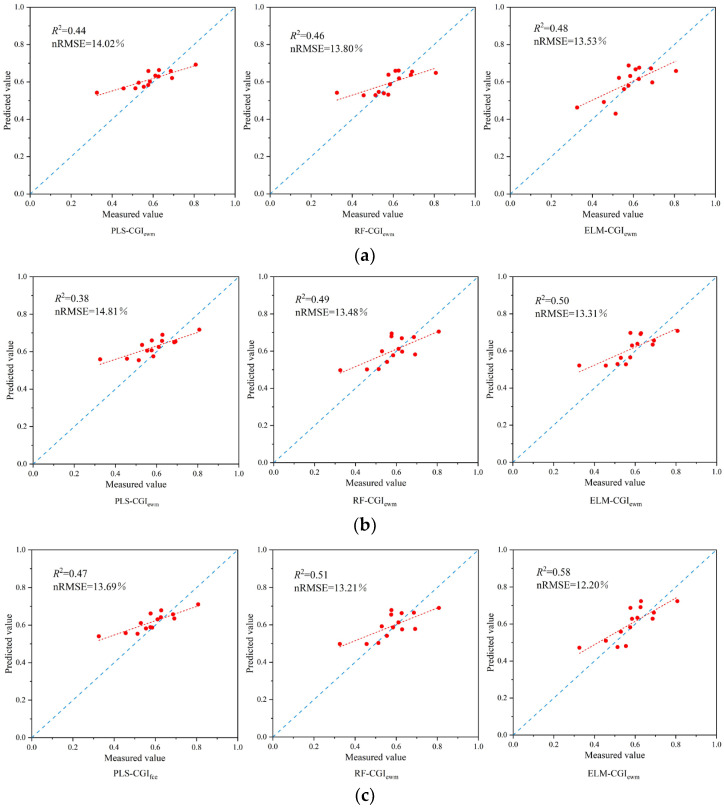
Accuracy verification of the CGI_ewm_ inversion model. (**a**) Model construction based on Vis. (**b**) Model construction based on TFs. (**c**) Model construction based on VIs and TFs.

**Figure 9 sensors-23-08089-f009:**
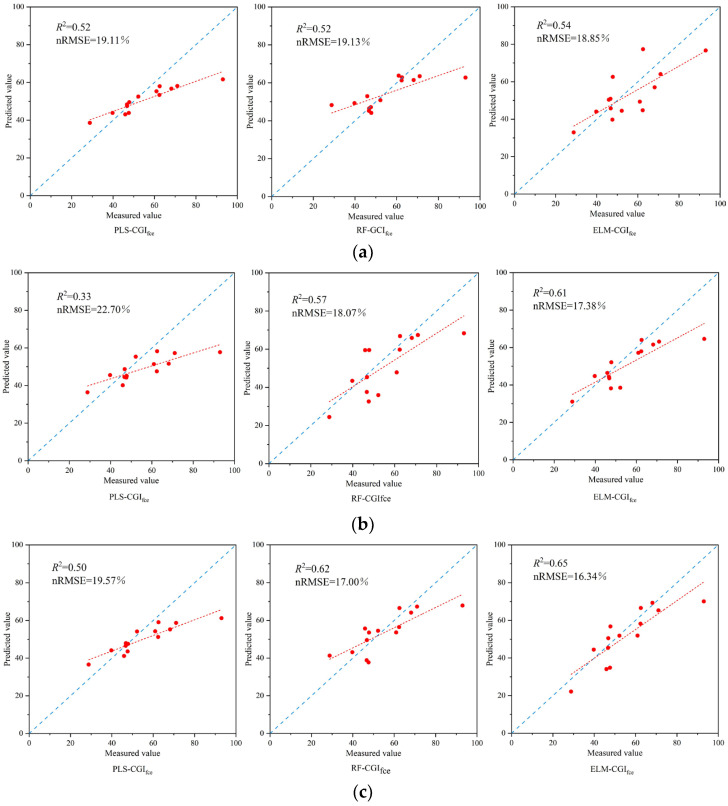
Accuracy verification of the CGI_fce_ inversion model. (**a**) Model construction based on Vis. (**b**) Model construction based on TFs. (**c**) Model construction based on VIs and TFs.

**Figure 10 sensors-23-08089-f010:**
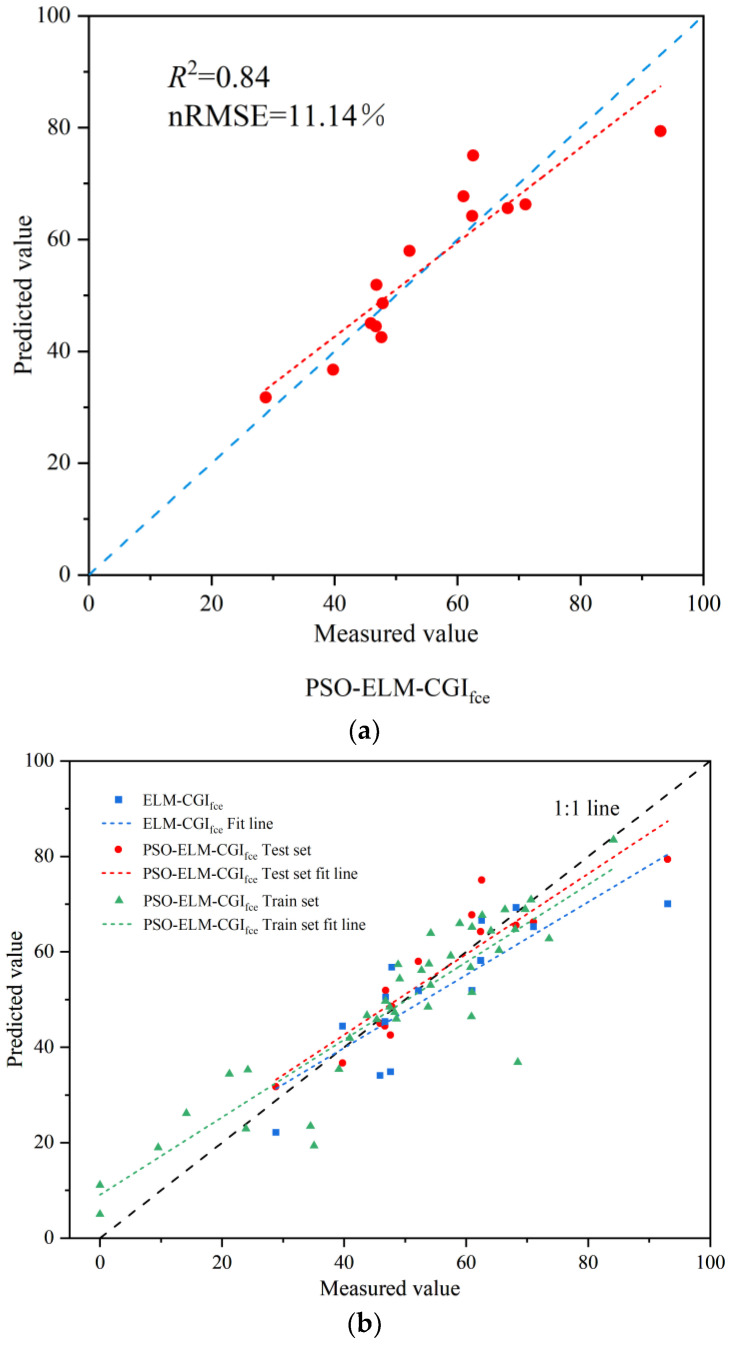
Inverse results of CGI_fce_ of PSO-ELM-CGI_fce_. (**a**) Validation of the PSO-ELM-CGIfce model. (**b**) Comparison of accuracy of the ELM-CGIfce model and PSO-ELM-CGIfce model.

**Table 1 sensors-23-08089-t001:** Multispectral Camera Parameters.

Band Name	Blue	Green	Red	Red Edge	Near Infrared
Center Wavelength (nm)	450	560	650	730	840
Band Width (nm)	32	32	32	32	52

**Table 2 sensors-23-08089-t002:** VIs and formula.

Vegetation Index	Abbreviations	Formula	Reference
Re-Normalized Vegetation Index	*RDVI*	*RDVI* = (*NIR* − *R*)/(*NIR* + *R*)^0.5	[27]
Ratio Vegetation Index	*RVI*	*RVI* = *NIR*/*R*	[28]
Normalized Difference Vegetation Index	*NDVI*	*NDVI* = (*NIR* − *R*)/(*NIR* + *R*)	[29]
Difference vegetation index	*DVI*	*DVI* = *NIR* − *R*	[30]
Soil-Adjusted Vegetation Index	*SAVI*	*SAVI* = 1.5(*NIR* − *R*)/(*NIR* + *R* + 0.5)	[31]
Optimized Soil-Adjusted Vegetation Index	*OSAVI*	*OSAVI* = 1.16(*NIR* − *R*)/(*NIR* + *R* + 0.16)	[32]
Triangular vegetation index	*TVI*	*TVI* = 60(*NIR* − *G*) − 100(R − G)	[33]
Excess Red Index	*EXR*	EXR = 1.4*R* − *G*	[34]
Normalized Difference Red Edge Index	*NDRE*	*NDRE* = (*NIR* − *RE*)/(*NIR* + *RE*)	[35]
Normalized Pigment Chlorophyll Index	*NPCI*	*NPCI* = (*R* − *B*)/(*R* + *B*)	[36]
Enhanced Vegetation Index2	*EVI*2	*EVI*2 = (*NIR* − *R*)/(1 + *NIR* + 2.4*R*)	[37]
Modified Vegetation Index	*MVI*	$MVI=\sqrt{\frac{NIR-R}{NIR+R}+0.5}$	[37]

Note: R, G, B, RE and NIR represent the reflectance of the red band, green band, blue band, red edge band and near-infrared band, respectively.

**Table 3 sensors-23-08089-t003:** TF and its calculation formula.

Parameters	Abbreviations	Formula
Mean	*mean*	$mean=\sum_{i,j}^{N-1}iP_{i,j}$
Variance	*var*	$var=\sum_{i,j=0}^{N-1}iP_{i,j}{\left(i-mean\right)}^{2}$
Contrast	*con*	$con=\sum_{i,j=0}^{N-1}iP_{i,j}{\left(i-j\right)}^{2}$
Dissimilarity	*dis*	$dis=\sum_{i,j=0}^{N-1}iP_{i,j}\left|i-j\right|$
Homogeneity	*hom*	$hom=\sum_{i,j=0}^{N-1}i\frac{P_{i,j}}{1+{\left(i-j\right)}^{2}}$
Entropy	*ent*	$ent=\sum_{i,j=0}^{N-1}iP_{i,j}\left(-\ln P_{i,j}\right)$
Second moment	*sm*	$sm=\sum_{i,j=0}^{N-1}iP_{i,j}^{2}$
Correlation	*corr*	$corr=\sum_{i,j=0}^{N-1}iP_{i,j}\left[\frac{\left(i-mean\right)\left(j-mean\right)}{\sqrt{var_{i}*var_{j}}}\right]$

Note: $P_{i,j}=\frac{V_{i,j}}{\sum_{i,j=0}^{N-1}V_{i,j}}$; *V_i_*_, *j*_ represents the brightness value of the pixel at the *i*th row and *j*th column; and *N* is the size of the moving window used to calculate texture features.

## Data Availability

Not applicable.

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
