# Peer review of "Construction of a Winter Wheat Comprehensive Growth Monitoring Index Based on a Fuzzy Degree Comprehensive Evaluation Model of Multispectral UAV Data"

_sensors, 2023, doi:10.3390/s23198089_

Round 1

Reviewer 1 Report

Dear colleagues,

I am sure your article and ideas may be useful not only for wheat monitoring but for wide audience. However, I have some proposals to improve the manuscript.

Title — please, check it carefully (model of UAV ?)

lines 148–153 — please, re-organize this information in the table form

table 2 — evidently you try to use the Pearson correlation, but it is for normally distributed data. Did you check normality of your data?

enthropy (eq 3) — In 1948, Claude Shannon suggested to estimate information entropy by the following equation:

H = - K Σ pi log pi

where K is a positive constant and pi is a probability.

He wrote that "the constant K merely amounts to a choice of a unit of measure". Commonly, K is not used. Generally speaking, H may vary between 0 and unlimited positive values, and if general number of samples increases H (information entropy) increases as well. This is fine for estimations of the entropy levels. Besides, generally speaking, if we will discuss information entropy, we should use preferably the base 2 and bits as the measurement unit.

Later, biologists and ecologists started to use this index (H = -  Σ pi log pi ) to value diversity and found that there were (and are) some problems with interpretation of the Shannon's index, because it depends of the object number (n) and the object distribution in the set of objects. The index increases (1) if the number of object increases and (2) if the object distributions become very similar. That means the maximum value of the Shannon index is log(n) when n is the total number of objects in the set.

To avoid this duality, in 1966, E. Pielou suggested to normalize the Shannon index (for the base e) and called this equation "evenness"

E =  H / Hmax =  (Σ pi ln pi) / ln n

this index varies between 0 and 1 (complete evenness), and there are no measurement units. Really this index is used as some measure of evenness per se, but not as a measure of entropy (the level of entropy is not limited). We often call this index the Pielou's evenness.

Please, explain the sense and distribution pattern of eq. 3

lines 384 and 578 — pearson > Pearson

Subsection 3.4.2 — it is not good practice to repeat the same values on figures and in the text.

Subsection 3.4 (and 5) — I am sure that accuracy/precision of your original data (AGB, SPAD, LWC, reflectance) is not very high (actually, at least, for biomass, perhaps, not more than several %%). This means accuracy of some values (R2 and nRMSE) is overestimated. I guess at least last two digits are excessive.

The title should be checked.

Author Response

Dear Reviewers/Editors:

Thank you very much for your review comments on the paper "Construction of Winter Wheat Comprehensive Growth Monitoring Index Based on Fuzzy Degree Comprehensive Evaluation Model of UAV Multi-Spectrum"(sensors-2584074). Each of your review comments has brought me important help in the revision of this paper and my future research work.

The main text has been revised according to the reviewer's comments, and the revised parts of the manuscript have been marked in red. In response to the reviewer's comments, I have made the following revisions:

For Reviewer #1

I am sure your article and ideas may be useful not only for wheat monitoring but for wide audience. However, I have some proposals to improve the manuscript.

(1) Title — please, check it carefully (model of UAV?).

Reply:

We are very grateful to the reviewers for their careful examination and patience in asking relevant questions.

We carefully checked the title of the paper and have now revised the title to "Construction of Winter Wheat Comprehensive Growth Monitoring Index Based on Fuzzy Degree Comprehensive Evaluation Model of Multispectral UAV Data". The title of the paper is marked in red.

(2) lines 148–153 — please, re-organize this information in the table form.

Reply:

Thanks for this comment. According to your comments, the content of this part has been appropriately deleted and adjusted and represented in the form of a table. Please refer to the Table 1 in Section 2.2.1.

(2) table 2 — evidently you try to use the Pearson correlation, but it is for normally distributed data. Did you check normality of your data?

 Reply:

Thanks for this comment. We checked the normality of the data. In this paper, we constructed comprehensive growth indicators based on biomass, chlorophyll content and water content of winter wheat, and used Pearson correlation analysis to calculate the correlation values between each growth indicator and the input characteristic parameters. Before the Pearson correlation analysis was performed, the data of winter wheat growth indicators were tested for normal distribution using the graphical method P-P plot of SPSS software. As shown in Figure 1, by the degree of conformity between the measured cumulative probability and the expected cumulative probability in the normal P-P plot of each growth indicator, it can be observed that the scatter distribution is approximately diagonal. In this way, we judged that the measured growth indicator data and the constructed data of the two comprehensive growth indicators conformed to the normal distribution law.

At the same time, we read a large number of related literatures before data processing. For example, Yue et al. used Pearson correlation to calculate the correlation between winter wheat biomass and vegetation index and texture features; Quan et al. used Pearson correlation to calculate the correlation between SPAD and spectral features.

Normal P-P plot of growth indicator data

The specific references are as follows:

a. Ghasemi, A.; Zahediasl, S. Normality tests for statistical analysis: a guide for non-statisticians. International journal of endocrinology and metabolism. 2012, 10, 486.

b. Öztuna, D.; Elhan, A.H.; Tüccar, E. Investigation of four different normality tests in terms of type 1 error rate and power under different distributions. Turkish Journal of Medical Sciences. 2006, 36, 171-176.

c. Gan, F. F.; Koehler, K.J.; Thompson, J.C. Probability plots and distribution curves for assessing the fit of probability models. The American Statistician. 1991, 45, 14-21.

d. Yue, J.B.; Yang, G.J.; Tian, Q.J.; Feng, H.K.; Xu, K.J.; Zhou, C.Q. Estimate of winter-wheat above-ground biomass based on UAV ultrahigh-ground-resolution image textures and vegetation indices. ISPRS Journal of Photogrammetry and Remote Sensing. 2019, 150, 226-244.

e. Quan, Y.; Zhang, Y.T.; Li, W.L.; Wang, J.J.; Wang, W.L.; Ahmad, I.; Zhou, G.S.; Huo, Z.Y. Estimation of Winter Wheat SPAD Values Based on UAV Multispectral Remote Sensing. Remote Sens, 2023, 15.14: 3595.

(3) enthropy (eq 3) — In 1948, Claude Shannon suggested to estimate information entropy by the following equation:

H = - K Σ pi log pi

where K is a positive constant and pi is a probability.

He wrote that "the constant K merely amounts to a choice of a unit of measure". Commonly, K is not used. Generally speaking, H may vary between 0 and unlimited positive values, and if general number of samples increases H (information entropy) increases as well. This is fine for estimations of the entropy levels. Besides, generally speaking, if we will discuss information entropy, we should use preferably the base 2 and bits as the measurement unit.

Later, biologists and ecologists started to use this index (H = - Σ pi log pi) to value diversity and found that there were (and are) some problems with interpretation of the Shannon's index, because it depends of the object number (n) and the object distribution in the set of objects. The index increases (1) if the number of object increases and (2) if the object distributions become very similar. That means the maximum value of the Shannon index is log(n) when n is the total number of objects in the set.

To avoid this duality, in 1966, E. Pielou suggested to normalize the Shannon index (for the base e) and called this equation "evenness"

E = H / Hmax = (Σ pi ln pi) / ln n

this index varies between 0 and 1 (complete evenness), and there are no measurement units. Really this index is used as some measure of evenness per se, but not as a measure of entropy (the level of entropy is not limited). We often call this index the Pielou's evenness.

Please, explain the sense and distribution pattern of eq. 3

Reply:

Thank you very much for your careful review and your patience in asking questions. In this paper, the entropy weight method was chosen to construct a comprehensive growth index of winter wheat by assigning weights to three single growth indicators of winter wheat biomass, chlorophyll and leaf water content. The size of the entropy weight value calculated by the entropy weight method reflects the amount of useful information of biomass, chlorophyll and leaf water content for the construction of the comprehensive growth index, and the smaller the information entropy and the larger the entropy weight of an index calculated, the greater the amount of information provided by the index. According to the size of the entropy measure the information utility value of the three growth indicators, so as to determine the weight of each indicator.

The calculation step is divided into three steps. Firstly, the original data matrix is standardized and normalized respectively, secondly, the entropy value of each indicator is calculated, and finally, the entropy weight value is calculated.

Equation 3 is the entropy value calculation formula in the second step of the entropy weight method.

where, ei denotes the information entropy of each longevity indicator of i. k is the standard coefficient, n is the number of indicators, and this study is three single indicators, so n=3;  is the value after standardization and normalization of the measured data of each indicator, and the larger the value is, it means that the uncertainty of the indicator is smaller, and the indicator is able to reflect the less information of the comprehensive growth situation of winter wheat.

We have added a note about the significance of Equation 3 at the relevant place in the paper. Please see the red words in Line 275~279.

 (4) lines 384 and 578 — pearson > Pearson

Reply:

Thank you very much for your careful review and your patience in asking questions. We have carefully checked and corrected lines 384 and 578 in the article and corrected "pearson" to "Pearson" in the whole article, and made corrections in the corresponding parts of the paper. Please see the red words in line 424 and line 621.

(5) Subsection 3.4.2 — it is not good practice to repeat the same values on figures and in the text.

Reply:

Thank you very much for the reviewer's valuable comments on my paper in your busy schedule. In response to your suggestions, we have adjusted the textual narrative in that part of the article, and tried to avoid the repetition of figures in the textual narrative and charts. Please see the red words in subsection 3.4.1 and subsection 3.4.2.

(6) Subsection 3.4 (and 5) — I am sure that accuracy/precision of your original data (AGB, SPAD, LWC, reflectance) is not very high (actually, at least, for biomass, perhaps, not more than several %%). This means accuracy of some values (R2 and nRMSE) is overestimated. I guess at least last two digits are excessive.

Reply:

Thank you very much for your careful examination and patience in asking questions. In response to your suggestion, we have rearranged R² and nRMSE to two decimal places in the figures of Subsection 3.4 (and Subsection 3.5) and in the text of the main text, as shown in Figure 8 and Figure 9. Please see the red words in the manuscript.

For Dear Editors

Thank you for arranging a timely review for our manuscript. We have revised our manuscript accordingly (see the red parts in the manuscript). If you still have any questions about this paper, please don’t hesitate to contact us.

Best wishes.

Mrs. Yu.

Reviewer 2 Report

In this work, the authors have established several machine learning models to realize the prediction of comprehensive growth monitoring index for winter wheat based on the multispectral images captured by UAV. Although the paper contains the necessary mathematical analysis and experimental verification, the work still has many details needed to be improved to provide convincing and clear enough results for readers. Some specific comments can be shown in the attached file.

Author Response

Dear Reviewers/Editors:

Thank you very much for your review comments on the paper "Construction of Winter Wheat Comprehensive Growth Monitoring Index Based on Fuzzy Degree Comprehensive Evaluation Model of UAV Multi-Spectrum"(sensors-2584074). Each of your review comments has brought me important help in the revision of this paper and my future research work.

The main text has been revised according to the reviewer's comments, and the revised parts of the manuscript have been marked in red. In response to the reviewer's comments, I have made the following revisions:

For Reviewer #2

(1) The authors have stated in the text “The pre-processed UAV multispectral images are acquired after the above processing with a resolution of 0.05 m.” (line 174 to line 175), and “An 18×18 (pixel) image centered on the sampling point and corresponding to the sample size and image resolution is cropped as the region of interest. The average reflectance of each band within each sample square is extracted.” (line 177 to line 179). Does it mean that the region of interest is with a size of 0.9 m × 0.9 m? However, the authors have also stated in the text “54 sample points are laid out in the area, each with a size of 1 m × 1 m. The sampling and testing methods for ground-based measurements are as follows.” (line 187 to line 188). It seems that the regions of interest with different sizes were used in the image pre-processing and ground data acquisition. Will it influence the experimental results?

Reply:

We would like to thank the reviewers for their careful examination and patience in raising the relevant questions. I would like to apologize for the incorrect statement of the resolution of the UAV in the paper.

The resolution of the UAV multispectral images acquired in this study is 0.05697 m. We have corrected the resolution in the paper to 0.05697 m. The size of the region of interest (ROI) was calculated based on the resolution of 0.05697 m. The determination of the size of the region of interest is described as follows:

Our field sampling consisted of two tasks: the acquisition of multispectral images from the drone and the acquisition of winter wheat growth indicators in the sample plots set up in advance. The size of the image region of interest (ROI) for the subsequent extraction of winter wheat reflectance in the sample plots from the UAV images was determined according to the size of the sample plots and the resolution of the images. The resolution of the UAV image acquired in this experiment was 0.05697m×0.05697m, and the sample size was set to 1m×1m, so the size of the region of interest was determined to be 1/0.05697, which was about 17.55 (pixel). The result of the calculation was retained to an integer during data processing as 18×18 (pixel), which is only 0.45 pixels different from 17.55 (pixel). This corresponds to the actual sample size of about 1.03×1.03 m, which is only 0.03m×0.03 m. The winter wheat was in the overwintering period at the time of this experiment, and the plants were widely spaced, so the difference of 0.03 m had almost zero effect on the plants sampled in the sample, and it did not have any effect on the reflectance extraction and the measured data, and it did not affect the results of the experiment. Please see the red words in line 179.

(2) The figure caption should be improved to accord with the content of the Figure 2. Besides, the authors have stated in the text “The camera is set to capture images at equal time intervals, with a time interval of 3 seconds, as shown in Figure 2.” (line 165 to line 166). However, such information cannot be found from Figure 2.

Reply:

Thank you very much to the reviewers for their careful examination and patience in asking relevant questions.

Figure 2 shows the field data collection site photos of UAV images. Therefore, based on the expert comments, we have restated the section and improved the caption of Figure 2. Please see the red words in line 169 and line 185.

(3) The authors have defined the global positioning system as GPS in line 187. However, GPS has appeared several times in the previous text (for example line 154, line 176). Some missing full names of abbreviations for specialized vocabularies in the manuscript should also be supplemented.

Reply:

Thank you very much to the reviewers for their careful examination and patience in asking relevant questions. We did so by carefully reviewing the full text. The full name of GPS was added to the first occurrence of GPS in the text, and all subsequent occurrences were indicated by abbreviations. And we added the full names of other abbreviations in the article: particle swarm optimization (PSO) in line 32; DJ-Innovations (DJI) in line 150; real time kinematic (RTK) in line 151; inertial measurement unit (IMU) in line 153. Please see the red words in the manuscript.

(4) The authors have stated in the text “The [0,1] is divided equally into 5 intervals corresponding to 5 rating levels. The fuzzy boundary gap between the head and tail is 0.15, and the center point gap is 0.1. The fuzzy boundary centers for adjacent rating levels are 0.2, 0.4, 0.6, and 0.8 respectively, as shown in Figure 3.” (line 280 to line 283). However, those specific values are not presented in Figure 3. Therefore, I suggest that the authors should improve the figure.

Reply:

Thank you very much to the reviewer expert teacher for raising this issue, we have improved and modified the figure according to the expert's opinion, as shown in the following figure. In the figure, (1) (2) (3) (4) (5) represent five grades; the interval between the head and tail fuzzy boundaries is the distance between a and h from the boundaries of 0 and 1 respectively, which is 0.15; the intermediate intervals are the intervals between a, b, c, d, e, f, g, and h, which is 0.1; and the center of fuzzy boundaries is 0.2, 0.4, 0.6, and 0.8. All of them have been marked in the figure with additional labels, as shown in Figure 3.

(5) The references list could be more complete and more correct. For example, the formula of OSAVI in Table 1 can be found as OSAVI= (NIRR)/(NIR+R+0.16) in Reference 32. However, it can be found it as OSAVI= 1.16(NIR- R)/(NIR+R+0.16) in Reference 34. Also, the formula of EXR in Table 1 can be found as EXR= (1.4G-R)/(B+G+R) in Reference 34. Therefore, the authors should improve the references list or explain the modification of the formulas cited.

Reply:

Thank you very much for your careful examination and patience in raising the relevant questions, and we apologize for the inappropriate citation of this part of the literature. In response to your question, we have carefully reviewed the literature cited in each vegetation index, and have corrected the citations of the vegetation index OSAVI and the citations of the EXR index. We have also checked and corrected the references of other vegetation indices. The relevant corrections are marked in red in the corresponding parts of the paper. Please see the red words in the Table 2 and lines 730~750.

(6) Figure 4 should be improved. Although VIs, TFs, CGIewm, and CGIfce are all pointed to be the input variables of the models in this paper, it is very hard to understand the figure. Why are there two parts named Variable Construction? Are there any differences between them? What are the meanings of the numbers 1, 2, and 3 in the figure? Why do the authors connect VIs, TFs, CGIewm, and CGIfce using such lines? Are VIs and TFs entered into the model twice?

Reply:

Thanks for this comment. There are two parts of the figure.4 named variable construction, one of which represents the extraction and construction of the characteristic variables and the other represents the construction of the inverse model of the composite indicator. The figure connects VIs, TFs, CGIewm, and CGIfce with lines labeled 1, 2, and 3, indicating that the two features, VIs, and TFs, are divided into three groups of input variables to construct the inversion models for CGIewm and CGIfce, respectively. The input variable groupings include the VIs group, the TFs group, and the combination of the two. So it is divided into three input models.

We have revised and improved Figure 4 to address the issues raised by the experts. The meanings of â‘ , â‘¡ and â‘¢ are also indicated. Specifically: " Note: AGB stands for above ground biomass, SPAD indicates Leaf chlorophyll content, LWC stands for leaf water content; â‘ : VIs are used as input variables. â‘¡: TFs are used as input variables. â‘¢: a combination of VIs and TFs is used as an input variable.". Please the see red words in the Figure 4 and lines 217~230.

(7) The authors should explain more about the values of CGIfce for Sample 3 and Sample 21 shown in Figure 5. Are they zero or missing?

Reply:

Thanks for this comment. The value of CGIfce for sample 3 and sample 21 in Figure 5 shows zero. It is not missing, but the value calculated by Eqs. (8)~(10) is zero.

In this study, CGIfce is a comprehensive growth index of winter wheat constructed by combining AGB, SPAD and LWC using the fuzzy degree evaluation modeling method. For the values of AGB, SPAD and LWC of each sample, the first is standardized; then each indicator is brought into the trapezoidal affiliation function, the function graph is shown in Figure 3, and its value in the five levels of trapezoidal function is calculated, and after the calculation is completed, the fuzzy degree matrix R of the sample is obtained; and then multiplied by the weight matrix W, the fuzzy vector B of the evaluation set of this sample is obtained; and finally, quantized as a percentage, that is, the comprehensive growth indicator of the modified sample is obtained. According to the calculation process and formula of the fuzzy comprehensive evaluation model selected in this paper, the CGIfce values of Sample 3 and Sample 21 are zero. The analysis of the results of the comprehensive index construction can be obtained that the smaller the CGIfce value is, indicating that the winter wheat growth is less affected by the ground collapse and damage, and the better the growth situation. The CGIfce value of Sample 3 and Sample 21 is zero, which indicates that the winter wheat in the sample is in good condition. And we have added to that section. Please see the red words in the lines 413~418.

(8) The authors have provided the software information for image pre-processing. I suggest that the authors should also supplement the software information and programming language for realizing the machine learning algorithms.

Reply:

Thank you very much for carefully reviewing and raising this issue. We have added the software information and programming language of the machine learning algorithms in the text based on your comments.

We used MATLAB 2022 software to program partial least squares (PLS), random forest (RF) and extreme learning machine (ELM) and particle swarm optimization extreme learning machine (PSO-ELM) regression algorithms, thus realizing the inverse modeling of integrated growth indexes of winter wheat. Please see the red words in the lines 334~336.

For Dear Editors

Thank you for arranging a timely review for our manuscript. We have revised our manuscript accordingly (see the red parts in the manuscript). If you still have any questions about this paper, please don’t hesitate to contact us.

Best wishes.

Mrs. Yu.

Reviewer 3 Report

Yu and coworkers have prepared a thorough paper describing their work to develop a comprehensive growth monitoring index based on fuzzy degree comprehensive evaluation model of multi-spectral UAV data. The paper is scientifically rigorous, and the claims well justified.  It advances the field by developing a comprehensive growth indicator that integrates texture indices along with the typical vegetative indices and imparts a non-uniform weighting scheme for the input variables.  In general, the paper is ready for publication after addressing the minor concerns noted below and a minor to moderate review of the English language. 

Minor concerns 

·        The title should be written as “Construction of Winter Wheat Comprehensive Growth Monitoring Index Based on Fuzzy Degree Comprehensive Evaluation Model of Multispectral UAV data” or something similar to improve the English language readability.

·        In Section 2.1. please include the country in the location description

·        Lines 138-140 contain a sentence that is not understandable.  This could be an English language problem.  Please review and modify.

·        Line 156; Sentence that starts “Conduct” should be replaced with “We conducted…”

·        Section 2.2.1; Please describe the calibration whiteboard in more detail (size, manufacturer) and the location within the Figure 1.

·        Section 2.2.2 Regarding the 20 representative plants selected from each sample plot… is a sample plot equivalent to one of the red circles in Figure 1?   Were twenty plants selected for each of the 54 sample areas?  Please clarify.

·        Section 2.3.2. Although most people obtain the texture features from grayscale images, the authors identified them from the grayscale images of two different color channels, R and G, if I understand correctly.  It would be good to include a direct statement of that here. Also, what is the bit-depth for the different channels collected?  8-bit or 16-bit?

·        Section 2.3.3 was the selection of 5 levels in the FCE arbitrary? What let to the decision to use 5 levels?

·        Figure 7 is unreadable in its present form.  Please enlarge a and B and place them vertically on top of one another so that the details can be red.  Also, in Figures 6 & 7 please add a description in the Figure caption of what the shape and orientation of the ovals mean.

·        Line 580 delete the word “the” in the phrase “and the most”

I think the paper needs a minor to moderate review of the English language.  I was able to follow the work and results with just a little extra effort, but I think an improvement to the English language would help readers who might e less familiar with the field understand it more easily. I made a suggestion for an improvement to the title.

Author Response

Dear Reviewers/Editors:

Thank you very much for your review comments on the paper "Construction of Winter Wheat Comprehensive Growth Monitoring Index Based on Fuzzy Degree Comprehensive Evaluation Model of UAV Multi-Spectrum"(sensors-2584074). Each of your review comments has brought me important help in the revision of this paper and my future research work.

The main text has been revised according to the reviewer's comments, and the revised parts of the manuscript have been marked in red. In response to the reviewer's comments, I have made the following revisions:

For Reviewer #3

Yu and coworkers have prepared a thorough paper describing their work to develop a comprehensive growth monitoring index based on fuzzy degree comprehensive evaluation model of multi-spectral UAS data. The paper is scientifically rigorous, and the claims well justified. It advances the field by developing a comprehensive growth indicator that integrates texture indices along with the typical vegetative indices and imparts a non-uniform weighting scheme for the input variables. In general, the paper is ready for publication after addressing the minor concerns noted below and a minor to moderate review of the English language.

Minor concerns:

(1) The title should be written as “Construction of Winter Wheat Comprehensive Growth Monitoring Index Based on Fuzzy Degree Comprehensive Evaluation Model of Multispectral UAV data” or something similar to improve the English language readability.

Reply:

Thank you very much for your comments. This paper has been revised to "Construction of Winter Wheat Comprehensive Growth Monitoring Index Based on Fuzzy Degree Comprehensive Evaluation Model of Multispectral UAV Data" according to your request. The title of the paper is marked in red.

(2) In Section 2.1. please include the country in the location description.

Reply:

Thank you very much for your comments. We have revised and improved the research area profile map of this paper according to your comments. Countries are indicated in the text description and in the study area overview map. Please see Figure 1 and the red words in the line 131.

(3) Lines 138-140 contain a sentence that is not understandable. This could be an English language problem. Please review and modify.

Reply:

Thank you very much for your comments. We have revised the sentence in lines 138-140.

Specifically modified as "After field research, the study area has some land collapse and land destruction due to long-term underground coal mining. Therefore, its elevation shows a gradual decline towards the center collapse area. In order to repair the ecological safety problems caused by coal mining, a series of land reclamation measures have been carried out in the collapsed area. " Please see the red words in the lines 140~145.

(4) Line 156; Sentence that starts “Conduct” should be replaced with “We conducted…”

Reply:

Thank you very much for your comments. We have modified “Conduct” to "We conducted".

The specific modification is "We conducted a survey of the geographical conditions of the study area and the surrounding structures to ensure avoidance of interference from objects such as high-voltage power lines and tall trees during the drone flight." Please see the red words in the line 158.

(5) Section 2.2.1; Please describe the calibration whiteboard in more detail (size, manufacturer) and the location within the Figure 1.

Reply:

Thank you very much for your comments. The whiteboard model used in this study is SRT-99-100 with a diffuse reflectance of 99% and a size of 25.4 cm × 25.4 cm. We have added detailed parameters of the radiometric calibration whiteboard to the article.

After the drone image data acquisition and before the drone returned, we placed the whiteboard on a flat, unobstructed surface in the study area. The UAV was manually controlled and hovering at about 2.5 m above the whiteboard to capture images of the whiteboard for radiometric correction in the subsequent UAV image preprocessing. The whiteboard did not appear in the drone footage of the study area during our operation.  Please see the red words in the lines 164~163.

(6) Section 2.2.2 Regarding the 20 representative plants selected from each sample plot… is a sample plot equivalent to one of the red circles in Figure 1? Were twenty plants selected for each of the 54 sample areas? Please clarify.

Reply:

Thank you very much for your comments. The 54 red points shown in the map of the study area indicate the GPS location points of the 54 sample areas laid out in this experiment, corresponding to the location of the sample center. In each 1*1m sample areas, 20 representative winter wheat plants were selected using the five-point sampling method. The plants were cut off at the ground level, separated from the leaves and stems, and then stored in sealed bags for subsequent determination of winter wheat biomass and other indicators. Therefore, the red dots in Figure 1 only represent the locations of the sample areas, and are not equivalent to the sample plots; and we have adopted the five-point sampling method for all 54 sample areas, and 20 representative plants were selected for each sample areas. We have redescribed this part of the article according to the expert opinion. Please see the red words in the lines 194~197.

(7) Section 2.3.2. Although most people obtain the texture features from grayscale images, the authors identified them from the grayscale images of two different color channels, R and G, if I understand correctly. It would be good to include a direct statement of that here. Also, what is the bit-depth for the different channels collected? 8-bit or 16-bit?

Reply:

Thank you very much for your comments. We have added the texture information extracted from the R band and G band images in this study based on the expert’s opinion and stated that the R band and G band are 8-bit depth. Please see the red words in the line 258~260.

(8) Section 2.3.3 was the selection of 5 levels in the FCE arbitrary? What let to the decision to use 5 levels?

Reply:

Thank you very much for your comments.

In this study, the choice of five levels in the fuzzy degree comprehensive evaluation model is not arbitrary. It was decided based on the actual growth condition of winter wheat and reference to related literature.

Entropy weight method and fuzzy comprehensive evaluation model were selected to complete the construction of comprehensive growth indicators of winter wheat. The entropy weight method is based on the entropy to calculate the weights of the indicators, and the entropy is the amount of effective information provided by each indicator. Fuzzy degree comprehensive evaluation model is a comprehensive evaluation method based on fuzzy degree math. The first step is to determine the factor vector and weight vector. In this study, the factor vector consists of three indicators: aboveground biomass, chlorophyll content and leaf water content, and the weight is the weight vector calculated by entropy weight method; the second step is to determine the evaluation level, which corresponds to this study both the biomass, chlorophyll content and water content of winter wheat are graded. It is divided into five grades, which represent poor growth, relatively poor growth, medium growth, relatively good growth and good growth of winter wheat. The third step is to determine the fuzzy degree matrix and affiliation function. The last step is to construct the fuzzy degree comprehensive evaluation model and assessment number.

Due to the long-term underground coal mining in this study area has resulted in ground collapse and damage, the elevation of the study area shows a gradual decline towards the center collapse area. Different reclamation measures such as land leveling and guest soil backfilling were carried out in the area as well as including unmanaged areas of slopes. The effects of different coal mining subsidence and reclamation measures on arable land changed the biochemical parameters of crops to a certain extent. After the field study, it was found that the growth status of winter wheat in the study area has a great difference. Based on the actual measured data of winter wheat biomass, chlorophyll content and leaf water content, and the actual situation of winter wheat growth in the study area, the winter wheat growth indexes were meticulously categorized into five grades. And we referred to the related literature, such as Xu et al. used the coefficient of variation method to establish the comprehensive growth index of winter wheat in the overwintering period, and selected the optimal model to carry out the inversion and mapping of the comprehensive growth index in the study area, and divided the comprehensive growth index of the whole area into five grades; Sun et al. used the fuzzy comprehensive evaluation model to construct the comprehensive evaluation index of maize collapse, where the evaluation grades were divided into five grades corresponding to the actual situation of the collapse in maize fields, respectively. The evaluation grades were categorized into five grades, corresponding to no, mild, moderate, severe and very severe corn fall, respectively.

In summary, there is a basis for setting the evaluation levels into five levels in the construction of the fuzzy degree comprehensive evaluation model in this study. We considered both the actual winter wheat growth situation and the research results of scholars. The specific references are shown below.

We have made additions accordingly. For details, please see the red words in the lines 304~307.

  1. Xu, Y.F.; Cheng, Q.; Wei X.P.; Yang, B.; Xia, S.S.; Rui, T.T.; Zhang, S.W. Monitoring of winter wheat growth under UAV using variation coefficient method and optimized neural network. T Chinese Soc Agric Eng. 2021, 37, 71-80.
  2. Sun, Q.; Chen, L.P.; Xu, X.B.; Gu, X.H.; Hu, X.Q. A new comprehensive index for monitoring maize lodging severity using UAV-based multi-spectral imagery. Comput Electron Agr. 2022, 202, 107362.

(9) Figure 7 is unreadable in its present form. Please enlarge a and B and place them vertically on top of one another so that the details can be red. Also, in Figures 6 & 7 please add a description in the Figure caption of what the shape and orientation of the ovals mean.

Reply:

Thank you very much for your comments. In response to the unreadability of Figure. 7, we have enlarged Figure. a and Figure b in Figure. 7, and have placed them vertically so that the details in the figures can be observed.

We have revised the titles of Figure 6 and Figure 7 to indicate meanings such as oval shape and direction. The specific modification is “Note: * indicates significant correlation at the 0.05 level, ** indicates significant correlation at the 0.01 level; Red color indicates positive correlation and blue color indicates negative correlation; right tilted ellipse indicates positive correlation and left tilted ellipse indicates negative correlation; flatter ellipse indicates higher correlation and wider ellipse indicates lower correlation.” Please see Figure. 7 and the red words in the Section 3.2 and 3.3.

(10) Line 580 delete the word “the” in the phrase “and the most

Reply:

Thank you very much for your comments. We have deleted the word “the” in the phrase “and the most”. The specific modification is “The correlation between CGIfce and most VIs is greater than that of CGIewm. ”. Please see the red words in the line 623.

For Dear Editors

Thank you for arranging a timely review for our manuscript. We have revised our manuscript accordingly (see the red parts in the manuscript). If you still have any questions about this paper, please don’t hesitate to contact us.

Best wishes.

Mrs. Yu.

Round 2

Reviewer 2 Report

The comments are answered in a suitable manner to achieve the requirement for publishing. Therefore, I suggest the acceptance of the manuscript in its present form.